# OpenReview forum: "How Easy is It to Fool Your Multimodal LLMs? An Empirical Analysis on Deceptive Prompt"
_NeurIPS.cc/2024/Workshop/SafeGenAi — SafeGenAi Poster_

### Official Review · Reviewer_zYFi · 2024-10-09
**This paper proposed an insightful benchmark but with limited innovation and exploration**

**Rating:** 5
**Confidence:** 3

**Review:**

This paper introduces MAD-Bench, a new benchmark for evaluating the robustness of multimodal large language models (MLLMs) against deceptive prompts. The authors curate 1000 image-prompt pairs across 5 categories of deception and evaluate 19 state-of-the-art MLLMs. The paper also proposes a simple remedy of adding an additional prompt paragraph to boost model performance.

Strengths:
- Evaluation: The authors evaluate a wide range of MLLMs, including both open-source and proprietary models, providing a broad view of the current state-of-the-art.
- Well-designed Benchmark: MAD-Bench covers multiple categories of deception, offering a nuanced assessment of model capabilities. The use of GPT-4o for automated evaluation, validated with human checks, is a practical approach for large-scale assessment.

Weaknesses:
- Limited Innovation: While the benchmark and evaluation are valuable, this analysis paper does not propose significant technical innovations in model architecture or training methods to address more about the identified issues.
- Benchmark Limitations: The benchmark, while diverse, is still limited to 1000 samples and 5 categories. A more extensive benchmark might reveal additional insights or challenges.
- Remedy Limitations: The proposed remedy, while effective, is relatively simple. Based on this analysis paper's research direction and positioning, a more comprehensive exploration of potential solutions is needed.

---

### Official Review · Reviewer_CrJY · 2024-10-09

**Rating:** 8
**Confidence:** 5

**Review:**

## Summary of paper
The authors present MAD-Bench, a benchmark of 1000 deceptive image/text pairs meant to test the understanding of MLLMs. The samples are divided into 5 main categories: count of object, non-existent object, object attribute, scene understanding, and text recognition. The authors evaluate 19 different open-source and proprietary models using this dataset, where GPT-4V performs the best. They then introduce a simple but effective performance booster which asks the model to think critically about the image.

## Strengths
- Well thought out categories.
- The benchmark will be useful for papers moving forward.
- They demonstrate benchmark using 19 different models.
- The authors provide a method to increase performance for models.

## Weaknesses
- N/A great job!

## Notes
- Can you bold the top performer for open sources as well in Table 2?
- Why do you think performance degrades with Phi-3-vision on count of Object?
- Reminds me of this paper: https://arxiv.org/pdf/2406.05948. Might be an interesting tie in.

---

### Official Review · Reviewer_QFfW · 2024-10-09
**Review on How Easy is It to Fool Your Multimodal LLMs? An Empirical Analysis on Deceptive Prompt**

**Rating:** 6
**Confidence:** 4

**Review:**

Strength:

The paper is well-structured and easy to follow, allowing readers to grasp the experiments and methodology without confusion. The experiments are thorough and leave no major questions unanswered.

Weakness:

Like the authors mentioned prior works like LRV and MME, have also somewhat tackled deceptive prompts. The novelty of the technical approach feels somewhat limited when they say that the inclusion of prompts starting with “What, How, Where,” adds an additional layer of challenge to the evaluation of MLLMs.

To strengthen the contribution, the authors could further highlight the unique insights gained from their dataset compared to prior work. Illustrating that the insights they mentioned can only be gained from the inclusion of prompts starting with “What, How, Where,” could bolster the argument for the novelty and importance of the study.

---

### Official Review · Reviewer_81Ms · 2024-10-11
**The benchmarking dataset seems quite interesting**

**Rating:** 7
**Confidence:** 3

**Review:**

The paper proposes a new benchmarking dataset to assess the trustworthiness of multi-modal LLMs (MLLMs). The key idea behind the benchmark is to focus on information that does not exist in the image and ask "what/when/how" style questions about it. The prompts are generated adversarially using another MLLM. While the approach is not terribly novel, I think it adds another nice dimension to constructing trust-related benchmarks. The approach also seems quite extensible so practitioners should be able to extend the dataset according to their needs. Overall, I think the paper would raise nice discussions in the workshop.